# Identifying Biomarkers for Accurate Detection of Stress

**DOI:** 10.3390/s22228703

**Published:** 2022-11-11

**Authors:** Kiran Jambhale, Smridhi Mahajan, Benjamin Rieland, Nilanjan Banerjee, Abhijit Dutt, Sai Praveen Kadiyala, Ramana Vinjamuri

**Affiliations:** Vinjamuri Lab, Department of Computer Science and Electrical Engineering, University of Maryland Baltimore County, Baltimore, MD 21250, USA

**Keywords:** substance use disorder, biomarkers, EDA, RespiBAN, machine learning

## Abstract

Substance use disorder (SUD) is a dangerous epidemic that develops out of recurrent use of alcohol and/or drugs and has the capability to severely damage one’s brain and behaviour. Stress is an established risk factor in SUD’s development of addiction and in reinstating drug seeking. Despite this expanding epidemic and the potential for its grave consequences, there are limited options available for management and treatment, as well as pharmacotherapies and psychosocial treatments. To this end, there is a need for new and improved devices dedicated to the detection, management, and treatment of SUD. In this paper, the negative effects of SUD-related stress were discussed, and based on that, a few significant biomarkers were selected from a set of eight features collected by a chest-worn device, RespiBAN Professional, on fifteen individuals. We used three machine learning classifiers on these optimal biomarkers to detect stress. Based on the accuracies, the best biomarkers to detect stress and those considered as features for classification were determined to be electrodermal activity (EDA), body temperature, and a chest-worn accelerometer. Additionally, the differences between mental stress and physical stress, as well as different administrations of meditation during the study, were identified and analysed. Challenges, implications, and applications were also discussed. In the near future, we aim to replicate the proposed methods in individuals with SUD.

## 1. Introduction

A rapidly growing epidemic afflicting 40.3 million adults (Substance Abuse and Mental Health Services Administration (SAMHSA) 2020 [1]) and having increased 6% from 2018 to 2019 [2], substance use disorder (SUD) is a dangerous disease with the capability of drastically affecting an individual’s brain and behavior. Additionally, the treatment of homeless individuals with SUD by the American Society of Addiction Medicine (ASAM) was incredibly challenging not only because these individuals dealt with immunocompromised systems, but also because of the COVID-19 pandemic [3]. According to the National Institute on Drug Abuse (NIDA), the rise in healthcare costs and job losses, which were especially augmented during the pandemic, resulted in the total expenditure of drug-related complications to exceed 500 billion dollars [4]. Even with the intense and expansive consequences of SUD, ranging from costs to the well-being of a large population, there are limited management and treatment options, pharmacotherapies [5], and psychosocial treatments available for SUD. The few devices addressing this issue, some examples being ReSET by Pear Therapeutics [6], A-CHESS [7], and NSS-2 Bridge by Massimo [8], mainly use cognitive behavioural therapy (CBT) [9] that are self-determined or used in accordance with a medical professional to guide patients on their next steps to managing SUD. While its integration into daily life and different formats make it effective for some, CBT generally falls short of achieving its goal in patients who lack the emotional strength to confront their addiction or are not mentally willing to incorporate changes into their current lifestyles. To address this epidemic, NIDA emphasized the importance of developing new and improved strategies to detect, treat, and manage SUD through their mission and strategic plan of improving individual and public health [4].

Currently, measurements of stress are usually conducted through self-reports [10], but this has several practical limitations as the symptoms reported can be exaggerated or under reported based on the patient’s mindset towards seeking help and their desire to be cleansed of substances. However, with the advent of wearable biomedical sensors in the form of smart watches or bracelets in everyday life [8], society has started to become more aware of and receptive to the integration of technology and health. The use of sensors in similar forms to measure physiological signals, such as electrodermal activity (EDA) [10,11,12] in the skin and electroencephalography (EEG) [13,14,15] from the scalp, is anticipated to be widely used. A greater incorporation of these sensors, particularly in everyday wearable technologies, is already evident through the recent upgrades made by Fitbit [14]. In a pilot study by Carreiro et al., a portable biosensor was embedded in smart watches that could continuously monitor changes in EDA, and thus, track stress levels over a period of time [10,16].

While there are gold standard electrochemical sensors to detect stress, EDA sensors have the unique ability to provide continuous and independent measurements of a subject’s stress and emotion levels with convenience and comfort. Since previous research indicates that there is a strong causal relationship between stress and emotion [17], EDA sensors are particularly useful instruments in determining SUD-related stress levels in patients struggling to manage related symptoms. However, the data from these sensors are only useful when the measurements are quick and accurate. This requirement makes using this data a grueling and challenging process, as EDA sensors typically generate enormous amounts of data and need tremendous computing power for subsequent processing. Therefore, it is crucial to identify and design efficient detection and estimation algorithms using machine learning and artificial intelligence. Through the identification of changing emotions, the varying instances of stress, although measured as acute throughout the experiments, will also be applicable to chronic stress in those with SUD. Additionally, it is possible that adding emotions into the analysis would allow for stress to be measured more accurately; however, more studies are required to measure the accuracy and reliability of this technique. Other physiological features, such as heart rate and respiration, also undergo significant changes when individuals experience stress. These features can be presented as good indicators for identifying stress.

When under stress, a series of psychological and physiological changes occur in the body as a result of the activation of the body’s sympathetic nervous system [4]. However, these changes are multiplied in those suffering from SUD. Additionally, the parasympathetic nervous system is unable to function properly and cannot regularly maintain homeostasis in the bodies of substance users. However, psychological changes tend to provide subjective results, since they cannot be well-tracked using external devices or treatments and are mainly measured through self-reports. Therefore, the optimal way to monitor SUD-related stress is by measuring physiological changes, such as an increase in heart rate, greater electrodermal activity, and higher rates of respiration, among others. Identifying the point of these changes will allow for the onset of SUD-related stress or relapse to be detected in an objective manner.

In collaboration with psychiatrists and clinicians, our group at UMBC is dedicated to researching and developing devices with the ability to swiftly and accurately detect, manage, and eventually treat SUD and its related symptoms. Our research draws on the results of the publicly available WESAD dataset [18], which contains measurements of physiological biomarkers tracked throughout various emotional states. The study induced two types of stress, mental and social, which are also induced by SUD, making this dataset a good choice. In order to identify the best combination of biomarkers that indicate stress, all of the features tracked in the WESAD dataset were collectively analyzed through a multi-way classification, reflecting the various emotional and stress levels that one may experience in daily life. Additionally, we have been able to train our models to identify whether a mental or social stressor is being experienced by the subject at a given moment. This differentiation will prove to be useful in the further stages of this project, as the type of stress can dictate which prevention or intervention method will be the most efficient and effective. Preliminary results from this work were published in [19].

In the following sections, this paper expands upon the experimental design through materials and methods used (Section 2), an analysis of the data collected, different classification methods, different features (Section 3), and how this information shapes our knowledge of what data can be used for better SUD detection, prevention, and treatment, along with future alternatives and solutions (Section 4).

## 2. Materials and Methods

This section describes the protocol of the original WESAD study (Section 2.1), pre-processing and preparation (Section 2.2), and the types of classifications used for different combinations of data (Section 2.3).

### 2.1. Experimental Protocol

The publicly available multimodal dataset for Wearable Stress and Affect Detection (WESAD) was used to study the physiological changes and responses to stress induced by SUD. These changes in the body were tracked using two wearable devices: RespiBAN Professional, which is worn around the chest, and Empatica E4, which is worn around the wrist. Embedded in these devices were sensors to track three axis accelerometers on the chest (X, Y, Z), electrodermal activity (EDA), electromyograph (EMG), respiration (RESP), electrocardiogram (ECG), and body temperature (TEMP); depicted in Figure 1. However, for the purpose of a computational analysis on identifying the features that would best indicate stress, only the recordings of RespiBAN Professional were used. This was chosen over Empatica E4 due to its larger volume of data points available (over 2 million data points per subject versus 20,000–50,000 data points per subject).

The original data collection was conducted for 17 subjects in total, but due to sensor malfunction, the data was only available for 15 subjects. Out of these remaining subjects, 12 were male and 3 were female, and they had a mean age of 27.5 ± 2.4 years. According to the study protocol, four emotional states—baseline, stress, amusement, and meditation—were induced in stages within all participants (Figure 2). Descriptions of these states and how they were induced are detailed below.

Baseline: A neutral state was induced as subjects sat or stood at a table and read neutral material.Stress: A highly strenuous state was induced in which subjects were exposed to both parts of the Trier Social Stress Test:
○Mental stress: a mental arithmetic task.○Social stress: a public speaking task.
Amusement: An amusing state was induced as subjects were shown funny video clips.Meditation: A de-excited state was induced as subjects were guided through meditation exercises.

At the end of each state, the participants filled out a questionnaire that asked them to rate different emotional states. The questionnaire for all states included the Positive and Negative Affect Schedule (PANAS), the State-Trait Anxiety Inventory (STAI), and Self-Assessment Manikins (SAM) tests. The stress state had an additional Short Stress State Questionnaire (SSSQ). The ratings assigned within each of these tests were utilized as a standard ground truth to assess the validity of the stress detection models proposed.

### 2.2. Preprocessing and Analysis

To perform the exploratory and predictive analyses on the data, we used MATLAB^®^ Machine Learning Toolbox and Python. The Python libraries used included pandas, sklearn, matplotlib, and numpy. Specifically, the sklearn library provided the tools to implement various algorithms. The integrated development environment (IDE) used was Jupyter Lab in combination with the Anaconda platform.

In order to preprocess the missing values in the dataset, values were first substituted based on the mean or mode of the distribution. Then, the data was normalized to have optimal distribution, so that no value drove the model’s performance in one direction or skewed the prediction. All values had equal weightage and statistical importance in the dataset as a result of this.

The physiological biomarkers that were the best indicators of accurate stress detection were identified from the dataset through logistic regression, linear regression, and principal component analysis (PCA). Additionally, sequential forward feature selection utilizing quadratic discriminant analysis was conducted for the purpose of a feature analysis, as presented in Table 1.

### 2.3. Classification

In our preliminary work [19], we performed three types of classification. They were as follows:(a)2-way: stress vs. amusement;(b)3-way: stress vs. amusement vs. meditation;(c)4-way: stress vs. amusement vs. meditation vs. baseline.

In contrast to our preliminary work [19], three different types of classification were performed in order to differentiate social stress and mental stress, and to also explore the effects of meditation on stress:(a)3-way: baseline vs. meditation before stress vs. meditation after stress;(b)3-way: baseline vs. social stress vs. mental stress;(c)6-way: baseline vs. social stress vs. mental stress vs. amusement vs. meditation before stress vs. meditation after stress.

These classifications for predictive analysis were performed using three approaches:(a)logistic regression;(b)decision trees;(c)XGBoost (gradient-boosted decision trees).

K-fold cross-validation was implemented and the results of the predictive analysis were measured using accuracy and area under the curve as performance metrics. While accuracy is a standardized and commonly-used metric, it is also crucial to meticulously calculate the accuracy that could not be achieved. This will help to minimize false positives. Hence, for binary classification, we also analyzed the area under the curve to check the degree of separation between true positives and false positives. The ratio of trained and test data split was 4:1.

Here, we have described the three types of classifications we performed and the rationale behind these classifications.

(a)3-way: baseline vs. meditation before stress vs. meditation after stress.

There are two versions of this classification, depending on the time of occurrence of rest. In the first version, the sequence of events is as follows:

Baseline -> meditation before stress -> rest -> meditation after stress.

For the second version, the sequence of events is as follows:

Baseline -> rest -> meditation before stress -> meditation after stress.

In the study, different subjects were subjected to different versions and no subject was subjected to both versions. The aim of this classification was to compare how meditation before and after stress affects the human body. For the first version, the technique to find the exact data point where the meditation before stress begins was by finding the data point for the last occurrence of amusement and first occurrence of stress. The data points in between these two conditions represent meditation before stress, or ‘Med I’, as shown in Figure 2. Meditation after stress was computed by locating the last data point for stress and then picking up the remaining data points for stress. Thereafter, the data points for both these types of meditation were combined with the data points for baseline to form a new dataset, which was then used for predictive analysis using the aforementioned algorithms and metrics in Python for each subject. To attain a better understanding of the features that contributed the most, a forward feature selection technique using quadratic discriminant analysis was implemented and the top three contributing features were extracted for each subject. These were then used as inputs to the predictive analysis algorithms used to study the results.

Similarly, the data points for meditation before and after stress for version 2 were extracted. To find the data points for meditation before stress, the last data point representing rest was located and the first data point representing amusement was located. The data points in between these represent meditation before stress for version 2, as shown in Figure 2. The data points representing meditation after the last occurrence of amusement were labeled as meditation after stress. These extracted data points were then combined with baseline data to form a new dataset, which was then subjected to predictive analysis using the above three algorithms for classification, and independently judged using accuracy and area under the curve as metrics for all subjects. Thereafter, using forward feature selection and quadratic discriminant analysis, the top three contributing features were extracted, and the same predictive analysis procedure was implemented.

(b)3-way: baseline vs. social stress vs. mental stress

All the subjects that participated in this experiment were subjected to two types of stress: mental stress and social stress. Mental stress was observed while computing a mathematical problem or counting down numbers, whereas social stress was observed during public speaking. The WESAD dataset represents these stress types with only one label. However, every subject was first subjected to social stress first and then mental stress. The duration of the two types of stress, and therefore, the number of data points for each of these stress types, was the same. This allowed us to correctly label the first half of the stress data points as social stress and the second half as mental stress. After extracting these data points, a new dataset was formed that consisted of baseline, social stress, and mental stress. This dataset was then used as an input to a predictive analysis model for classification using logistic regression, decision trees, and XG-Boost algorithms, and then this performance was judged using accuracy and area under the curve as the classification metrics for each subject. Furthermore, using forward feature selection and quadratic discriminant analysis, the top three features were extracted from the above-formed dataset and then independently subjected to the same classification for every subject as above.

(c)6—way: baseline vs. social stress vs. mental stress vs. amusement vs. meditation before stress vs. meditation after stress

For this predictive model, we combined the above two datasets in accordance with the versions that the subjects were exposed to. Additionally, the amusement labels were also appended to the dataset. The sequence of events for the first type of classification (or version 1) are as follows:

Baseline -> Amusement -> Meditation before stress -> Social Stress -> Mental Stress -> Rest -> Meditation after stress.

The sequence of events for the second version are as follows:

Baseline -> Social Stress -> Mental Stress -> Rest -> Meditation 1 -> Amusement -> Meditation 2.

First, a simple classification was performed using predictive analysis that was implemented using logistic regression, decision trees, and XG-Boost algorithms, and then the performance was measured using accuracy and area under the curve for each subject separately. Thereafter, the top three contributing features were extracted using forward feature selection and quadratic discriminant analysis. Again, these features were used by the same three algorithms and measured using the same classification metrics.

## 3. Results

In this section, the analysis for choosing the top biomarkers, as well as the biomarkers themselves, were identified. The accuracies produced by each statistical test and combination of features were specified and accordingly ordered.

To calculate the most significant combination of physiological features, our first step was to implement sequential forward feature selection using quadratic discriminant analysis. For every candidate, we assessed which would be the most significant 2, 3, 4, 5, 6, and 7 features. Table 1 depicts the findings from this analysis, as well as all the features plotted against the corresponding candidates. The numerical value signifies the number of times the features were chosen in combination with other features for the same subject. On a scale of zero through six, zero implies the non-existence of that feature in any combination conducted for that subject (least relevant), while six indicates the occurrence of that feature in all combinations carried out for that subject (most relevant). A total of six iterations were performed with various feature counts ranging from a combination of two features to a combination of eight features. This selection process was different for different subjects.

We observed that the most used and considered feature in every combination for all but one of the 15 subjects was the accelerometer Z-axis (Z), due to its direct association to heartbeat. The feature that was second-most used and included in all six cases for four of the candidates was EDA. On the other end of the spectrum, the least used and the least significant feature used in none of the combinations for 13 subjects was the accelerometer X-axis (X). However, since the accelerometer, as a whole, contributed to the classification accuracy, this variable cannot be ignored for any of the future studies. Furthermore, to identify and validate features that could best predict stress, a feature analysis with logistic regression and PCA was performed for every subject. This helped us discover several other features that emerged as being more important than the rest. The features that had a strong association to a subjects’ mental and emotional states were EDA and temperature, along with the accelerometer (z-axis). The outcomes attained from logistic regression with 2- (stress vs. amusement), 3- (stress vs. amusement vs. meditation), and 4-way (stress vs. amusement vs. meditation vs. baseline) multivariate classification are displayed in Table 2. In general, logistic regression had an average accuracy (ACC) of 0.969 and an average AUC-ROC of 0.985 in all the subjects.

In addition to logistic regression, similar classification was repeated using the Decision Tree method. Not only did the Decision Tree choose the same features discovered from the logistic regression (EDA, temperature, and the accelerometer z-axis), but it also worked slightly better, with an average AUC-ROC of 0.998 and an average accuracy of 0.968. To validate the results, another classification algorithm, XGBoost, was used. This classifier also chose the EDA, temperature, and the accelerometer z-axis features, and it outperformed both the logistic regression and the Decision Tree classification algorithms in the classification accuracies. In all, a total of 135 individual tests were run—45 test runs with each of the 3 classifiers. Of these 135 tests, EDA was chosen as one of the top three features 121 times, making it the most significant feature. The second-most significant feature was temperature, as it was chosen 106 times as a top feature. Finally, the accelerometer z-axis was chosen as the third-most significant feature, due to being one of the top three features 76 times. These features, along with others, are depicted in Figure 3. Detecting whether the subject is stressed depends on a combination of various factors, so finding a relevant combination of pertinent features was a vital task. Selecting only one feature would probably not provide relevant and generalizable results.

In summary, we found that the predictive models performed better when supplied with the top three contributing features as inputs, instead of all the features. Logistic regression yielded lower accuracy and AUC score as compared to Decision Trees and XG-Boost. XG-Boost showed an overall excellent performance in most of the subjects. We were able to successfully achieve high performance even after considering six types of labels (6-way classification) by selecting the best combination of parameters for the predictive analysis models. In our limited experience with the data, the accelerometer was the most common feature in all subjects for all three types of classifications. Furthermore, we were able to effectively distinguish between mental stress and social stress, and thereby, create two new sub-labels which will be very useful in future studies.

Table 3 shows accuracies and AUC for all types of classifications for all subjects using three algorithms and all features:

Table 4 shows accuracies and AUC for all types of classifications for all subjects using three algorithms and the top three features:

## 4. Discussion

### 4.1. Optimal Biomarkers for Detection of Stress

Our objective in this paper was to identify optimal biomarkers that can best assist in the accurate detection of stress. We found that EDA, body temperature, and chest-worn accelerometers are important features in stress detection, as corroborated by other related studies [20,21,22]. We aimed not to find just one best feature, but a combination of multiple features that can help in precise detection as the fusion of multimodal features improves the detection of accuracy [22]. As shown in Table 2, optimal features change from subject to subject, and this enabled us to build personalized models that were unique to individuals. Additionally, we were able to train our models to identify which type of stressor, mental or social, was being experienced based on the various administrations of the experiment. This analysis can help researchers better identify the triggers for relapse and recommend informed and appropriate treatments.

Our study builds on research already conducted by using data provided through a multi-way classification to identify the fusion of sensors that would provide the best indication of stress. We also analysed and identified the stressor type, but we only limited it to mental and social stress, since physical stress does not fit our goal of treating and managing stress induced by SUD. Additionally, our group used various administrations of the tests, with varying orders of states in which the subjects were in, which allows for us to identify which states the person must be in order for the classifier to most accurately identify stress. Together, these goals allowed our group to focus on the aspects most prevalent in SUD-related stress and information potentially useful for treatment.

The WESAD dataset, having been openly available online since 2018, has been used in many other studies with similar goals—the detection of stress. Research closest to the one that our group conducted involved the identification of stress and stressor type through various machine learning classifiers and models. A study conducted by Iqbal et al. [20] conducted a two-way classification of the baseline and stressed states. Since the dependent variable of the state of the participants was binary, a logistic regression model was used over a linear regression model. This model identified respiration rate as the strongest indicator of stress, along with the combination of respiration rate, heart rate, and heart rate variability, providing an accuracy of 85.70%. Another study conducted by Elzeiny et al. [21] used the interbeat interval and blood volume pulse features from the photoplethysmography (PPG) with convolutional neural network (CNN) to achieve a stress detection rate of 98.10%, and average pixel intensity data to achieve an accuracy of 99.18%. Additionally, the type of stressor, categorized into physical, cognitive, and social, achieved an identification accuracy of 98.5% using CNN and 96.5% using extra trees.

### 4.2. Significance and Rationale for Detection of Stress in SUD and Other Disorders

Even though our aim was to detect stress as a trigger in the context of substance abuse, the methods outlined and developed in this paper have several applications in various domains, as stress can lead to multiple other complications. Stress negatively affects cognitive functions, weakens memory, increases blood pressure, and causes cardiac disorders and diabetes, to list a few, especially as instances of acute stress begin to build up in degree and instance, and can eventually develop into chronic stress among other, even more dangerous illnesses [17,23,24]. In each of these diseases and disorders, stress manifests in different ways, reiterating the benefit of the multimodal fusion of features in improving the detection accuracy across domains.

Decades of research have shown that stress increases risk of substance abuse, and could be a hindrance to effective treatment of substance abuse [25,26,27]. The use of substances is known to stimulate the release of the neurotransmitter dopamine, providing an intense pleasurable feeling which creates a positive feedback loop within the user. This leads to the uncontrolled use of illicit drugs, alcohol, excessive use of legal drugs, or other addictive behaviours [28]. Further negative effects of SUD can manifest themselves in substance users in the form of stress, among other symptoms. The detrimental cycle of addiction has the power to significantly impair the lives of users in terms of their decision-making skills, ability to meet responsibilities at school and/or work, personal relationships, and internal well-being. One’s physical dependence on a substance to get through these daily activities and experiences makes stepping away from a certain substance psychologically stressful. This may lead to relapse or Post-Acute Withdrawal Syndrome [3], a prolonged experience of withdrawal symptoms. When these instances of stress begin to add up or when their symptoms are prolonged, the acute stress measured in the experiment can develop into chronic stress. Additionally, any instance of acute stress can provoke a heart attack or a stroke. This connection between the two forms of stress provides for a better analysis for stress detection and management in substance users, since the physiological symptoms are likely to appear in the same way.

### 4.3. Limitations and Future Directions

The small number of participants (15 subjects) in this study may make the results appear less significant. In future, this will be addressed by repeating the WESAD administration with a greater number of participants; in particular, in individuals with SUD. However, as observed in the data analysis section, each individual presented a diverse set of biomarkers. The data analysis outlined in this paper can also be expanded to predict different levels of stress based on the scores that the subjects assigned to different emotional states. Additionally, since the test was administered in two different orders of emotional states, the correlation between stress levels before and after the meditation period can be examined, as this can also improve management and treatment options for SUD. Furthermore, the analysis in this paper considers personalized models with the data being trained for each individual participant. These specific models provide greater accuracy in the prediction of stress, as the effects of the various mental and social tasks are experienced and handled in different ways by different people, as evidenced by the questionnaire data. However, the information from the questionnaires was not included in the analysis, as our goal was to keep the method of prediction as objective as possible and not let the perception of stress be influenced by the individual. In future studies, we will consider measures to tightly tie the questionnaires in with the experimental biomarkers. It is important to acknowledge that the non-invasive neurophysiological features considered in this study for stress detection could be modulated due to other disorders unrelated to stress or SUD, thus confounding prediction of stress. In such circumstances, pharmacological interventions to cross validate may be deemed necessary. Lastly, the algorithms and methods proposed in this paper need to be validated in individuals with SUD. The methods proposed here serve only as a preliminary demonstration of proof of concept. Because we depended on a publicly available dataset, we had limited information about the study participants, nor did we have any information about the inclusion and exclusion criteria. Nevertheless, with the encouraging results obtained in this paper, we will repeat similar studies in individuals with SUD in the near future.

### 4.4. Towards Development of a Wearable Device for Detection and Management of Stress in SUD

Despite the scientific and technological advances in the areas related to brain and behaviour, there are limited devices and interventions available to help individuals with SUD. The goal of our multidisciplinary team is to develop an integrated portable system that is capable of detection and management of SUD using wearable biosensors, including EDA and EEG sensors. This system will have four major components: (1) based on well documented negative effects of stress in SUD [25,26,27], detection of stress and emotional states recorded using EEG and EDA sensors; (2) the use of machine-learning (ML) and artificial intelligence (AI) algorithms for improving the detection of stress, emotion, and behaviour; (3) provide neurofeedback using EEG sensors to measure, manage, and modify brain activity, and thus, associated behaviour; and (4) develop a smartphone app that can provide a user-friendly and personalized graphical user interface. The proposed technology, with its unique ability to influence the brain and behaviour, will impact individuals with this serious condition in a more immediate and personal manner. In collaboration with psychiatrists and clinicians, our group at UMBC is in the preliminary stages of developing a wearable device that can help in the detection, management, and eventual treatment of SUD in an efficient and effective way.

## 5. Conclusions

In an attempt to find better ways to address the detrimental effects of stress elicited by SUD, this study researched relevant biomarkers in the form of physiological signals tracked by RespiBAN, a chest-worn device. Extensive data analysis indicated that EDA, body temperature, and chest-worn accelerometer contributed the most to the accurate detection and classification of stress, along with other emotional states. We also separately analysed the detection of mental and social stress in the case of different situational triggers, as well as the first and second meditation states. In the near future, we hope to further our analysis of the type of stress and current emotional states, and how these pertain to better forms of treatment for individuals with SUD. As EDA can be detected across various physiological disorders unrelated to SUD, we will consider multimodal features for accurate detection of stress in SUD. We will also look at unique optimal physiological features in individuals to customize detection models. Although this paper highlights detecting stress and other emotional states in different contexts, we aim to research how to detect different levels of stress itself. We aim to test the proposed methods in individuals with SUD soon.

## Figures and Tables

**Figure 1 sensors-22-08703-f001:**
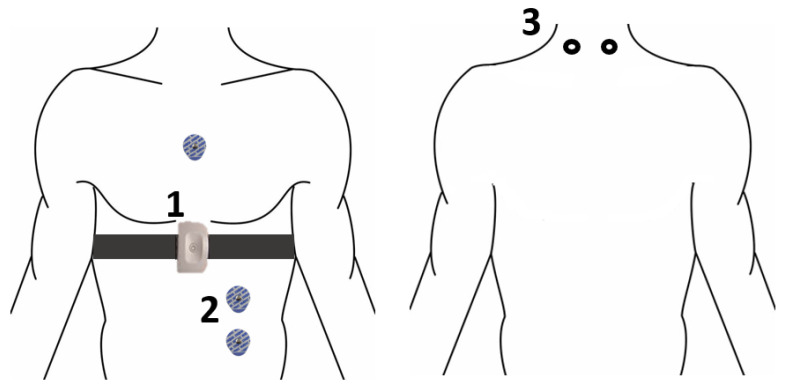
RespiBAN Professional’s placement of electrodes. 1. RespiBAN Professional with temperature, EDA, and control module. 2. Three ECG electrodes. 3. Two EMG electrodes on the back where the shoulder meets the neck. Adapted from [19].

**Figure 2 sensors-22-08703-f002:**
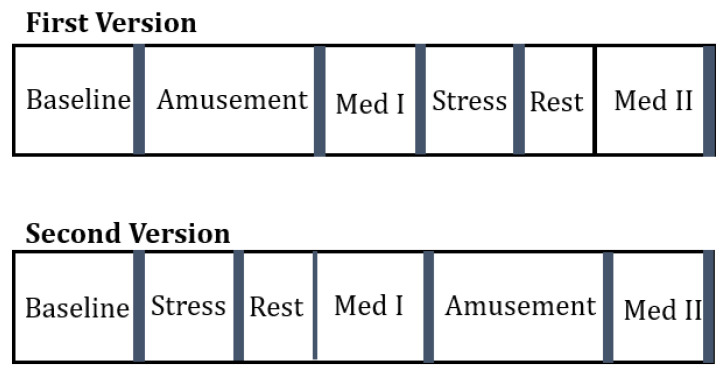
The two protocols tested under this study. The blue bars indicate the times when the study participants filled out questionnaires for self-report. Adapted from [18].

**Figure 3 sensors-22-08703-f003:**
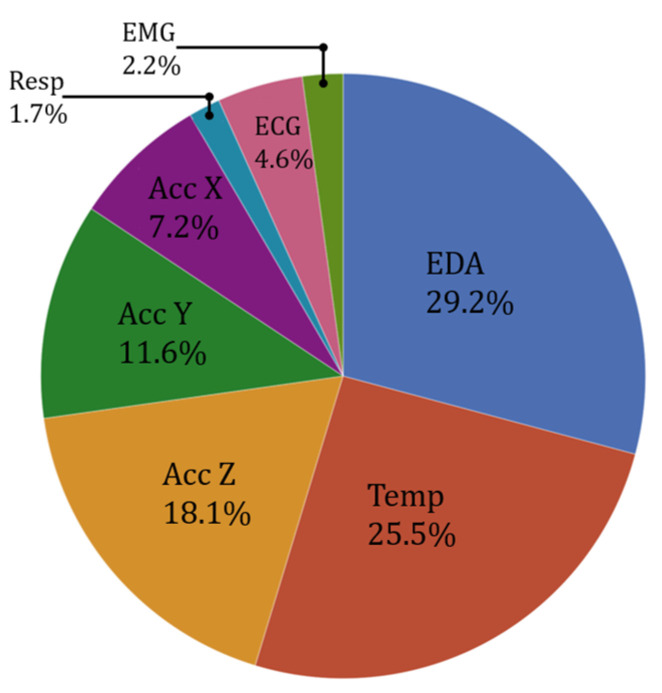
Significance of the features based on how frequently they were used in three different classifiers for best accuracies. EDA, temperature, and accelerometer Z (and Y) stand out as the important features.

**Table 1 sensors-22-08703-t001:** Significance of the features.

	ECG	EDA	EMG	Resp	Temp	X	Y	Z
S2	2	5	1	3	6	0	4	6
S3	2	5	3	1	6	0	4	6
S4	2	4	3	1	6	0	5	6
S5	2	6	3	1	4	0	5	6
S6	2	6	1	3	4	0	5	6
S7	1	5	2	3	6	0	4	6
S8	1	5	3	2	4	0	6	6
S9	2	6	1	3	5	0	4	6
S10	2	6	1	3	4	0	5	6
S11	0	4	1	3	5	2	6	6
S13	1	6	3	2	4	0	5	6
S14	2	4	1	3	5	0	6	6
S15	2	6	1	3	0	4	5	5
S16	1	6	2	3	4	0	5	6
S17	3	6	4	2	5	0	4	6

**Table 2 sensors-22-08703-t002:** Classification accuracy using logistic regression with top three features.

		AUC-ROC	ACC	Top 3 Features							
				ECG	EDA	EMG	Resp	Temp	X	Y	Z
S2	2-way	1	1		X			X		X	
	3-way	1	0.999		X			X			X
	4-way	0.999	0.997	X	X						X
S3	2-way	0.89	0.82	X	X			X			
	3-way	0.945	0.949		X			X			X
	4-way	0.965	0.886					X		X	X
S4	2-way	1	1		X					X	X
	3-way	0.999	0.996		X			X			X
	4-way	0.999	0.994		X			X	X		
S5	2-way	1	1		X					X	
	3-way	0.999	0.996		X			X			X
	4-way	0.999	0.994		X			X	X		X
S6	2-way	0.997	1		X			X		X	
	3-way	0.982	0.996					X		X	
	4-way	0.948	0.994		X			X			X
S7	2-way	1	1	X	X	X					X
	3-way	0.999	0.995		X			X		X	
	4-way	0.947	0.803		X			X			
S8	2-way	0.999	0.998							X	X
	3-way	0.999	0.996					X		X	X
	4-way	0.999	0.999					X		X	X
S9	2-way	0.993	0.982		X			X			X
	3-way	0.995	0.981		X			X	X		X
	4-way	0.982	0.971		X			X	X		
S10	2-way	1	1	X	X		X				
	3-way	0.999	0.999		X			X			
	4-way	0.999	0.997		X			X			X
S11	2-way	1	1					X	X	X	X
	3-way	0.997	0.971		X			X			
	4-way	0.988	0.929		X			X			X
S13	2-way	1	1	X	X	X					X
	3-way	0.998	0.981		X			X		X	
	4-way	0.998	0.981		X			X		X	
S14	2-way	0.919	0.903		X			X	X		
	3-way	0.873	0.819					X			X
	4-way	0.932	0.886		X			X		X	X
S15	2-way	1	1		X				X		
	3-way	0.999	0.998		X			X	X	X	
	4-way	0.998	0.992		X			X	X		
S16	2-way	1	1		X	X	X				
	3-way	0.999	0.998		X			X			X
	4-way	0.999	0.998		X			X			X
S17	2-way	1	1		X			X	X		
	3-way	0.999	0.995		X			X			X
	4-way	0.999	0.997		X			X			X

**Table 3 sensors-22-08703-t003:** Classification accuracy using all features.

		Accuracy			AUC		
		Logistic Regression	Decision Trees	XG-Boost	Logistic Regression	Decision Trees	XG-Boost
S2	Med 1 vs. Med 2	0.999	1.0	0.999	0.999	1.0	0.999
	Social vs. mental stress	0.967	0.988	0.933	0.991	0.998	0.999
	6-way	0.972	0.993	0.999	0.996	0.999	0.999
S3	Med 1 vs. Med 2	0.986	0.999	0.999	0.998	0.999	0.998
	Social vs. mental stress	0.937	0.996	0.999	0.950	0.999	0.999
	6-way	0.628	0.983	0.998	0.842	0.998	0.999
S4	Med 1 vs. Med 2	0.999	1.0	1.0	0.999	0.999	0.999
	Social vs. mental stress	0.989	0.993	0.995	0.999	0.999	0.999
	6-way	0.984	0.993	0.996	0.997	0.999	0.999
S5	Med 1 vs. Med 2	0.980	0.999	0.997	0.997	0.997	0.997
	Social vs. mental stress	0.928	0.979	0.988	0.957	0.990	0.997
	6-way	0.743	0.983	0.990	0.939	0.996	0.999
S6	Med 1 vs. Med 2	0.884	1.0	0.999	0.963	1.0	0.999
	Social vs. mental stress	0.979	0.995	0.999	0.957	0.990	0.997
	6-way	0.642	0.986	0.995	0.923	0.998	0.999
S7	Med 1 vs. Med 2	0.999	0.999	0.999	0.999	0.999	0.999
	Social vs. mental stress	0.995	0.997	0.998	0.999	0.999	0.999
	6-way	0.918	0.998	0.998	0.992	0.999	0.999
S8	Med 1 vs. Med 2	0.999	1.0	0.999	1.0	1.0	1.0
	Social vs. mental stress	0.912	0.973	0.983	0.955	0.994	0.998
	6-way	0.943	0.9821	0.989	0.99	0.998	0.999
S9	Med 1 vs. Med 2	1.0	1.0	0.999	1.0	1.0	0.999
	Social vs. mental stress	0.982	0.992	0.994	0.998	0.999	0.999
	6-way	0.973	0.989	0.995	0.996	0.999	0.999
S10	Med 1 vs. Med 2	1.0	1.0	1.0	1.0	1.0	1.0
	Social vs. mental stress	0.888	0.971	0.979	0.928	0.991	0.997
	6-way	0.875	0.999	0.999	0.951	0.999	0.999
S11	Med 1 vs. Med 2	0.875	0.999	0.999	0.951	0.999	0.999
	Social vs. mental stress	0.947	0.988	0.988	0.993	0.999	0.999
	6-way	0.858	0.977	0.985	0.981	0.999	0.999
S13	Med 1 vs. Med 2	0.999	1.0	0.999	0.999	0.999	0.999
	Social vs. mental stress	0.864	0.983	0.994	0.888	0.995	0.999
	6-way	0.907	0.984	0.994	0.966	0.997	0.999
S14	Med 1 vs. Med 2	0.971	0.999	0.999	0.995	0.999	0.999
	Social vs. mental stress	0.963	0.992	0.995	0.972	0.999	0.999
	6-way	0.841	0.967	0.981	0.973	0.992	0.999
S15	Med 1 vs. Med 2	1.0	1.0	0.999	1.0	1.0	1.0
	Social vs. mental stress	0.991	0.998	0.998	0.997	0.999	0.999
	6-way	0.991	0.997	0.999	0.999	0.999	0.999
S16	Med 1 vs. Med 2	0.923	0.997	0.999	0.963	0.999	0.999
	Social vs. mental stress	0.992	0.993	0.994	0.999	0.999	0.999
	6-way	0.934	0.989	0.994	0.989	0.999	0.999
S17	Med 1 vs. Med 2	1.0	1.0	0.999	1.0	1.0	1.0
	Social vs. mental stress	0.947	0.982	0.989	0.994	0.997	0.999
	6-way	0.956	0.980	0.993	0.994	0.997	0.999

**Table 4 sensors-22-08703-t004:** Classification accuracy using the top three features.

		Accuracy			AUC		
		Logistic Regression	Decision Trees	XG-Boost	Logistic Regression	Decision Trees	XG-Boost
S2	Med 1 vs. Med 2	0.999	1.0	0.999	0.999	1.0	0.999
	Social vs. mental stress	0.966	0.988	0.988	0.991	0.998	0.991
	6-way	0.979	0.992	0.993	0.998	0.999	0.998
S3	Med 1 vs. Med 2	0.997	0.999	0.998	0.999	0.999	0.999
	Social vs. mental stress	0.936	0.996	0.998	0.974	0.999	0.974
	6-way	0.720	0.983	0.993	0.912	0.998	0.912
S4	Med 1 vs. Med 2	0.999	1.0	1.0	0.999	0.999	0.999
	Social vs. mental stress	0.990	0.991	0.991	0.997	0.997	0.997
	6-way	0.964	0.988	0.985	0.993	0.999	0.993
S5	Med 1 vs. Med 2	0.988	0.999	0.997	0.998	0.998	0.998
	Social vs. mental stress	0.967	0.974	0.972	0.988	0.991	0.988
	6-way	0.24	0.959	0.971	0.973	0.993	0.973
S6	Med 1 vs. Med 2	0.995	1.0	0.998	0.999	1.0	0.999
	Social vs. mental stress	0.978	0.995	0.996	0.996	0.999	0.996
	6-way	0.629	0.979	0.977	0.941	0.998	0.941
S7	Med 1 vs. Med 2	0.999	0.999	0.999	0.999	0.999	0.999
	Social vs. mental stress	0.996	0.997	0.997	0.999	0.999	0.999
	6-way	0.946	0.998	0.997	0.995	0.999	0.995
S8	Med 1 vs. Med 2	0.999	1.0	0.999	0.999	0.999	0.999
	Social vs. mental stress	0.920	0.975	0.973	0.971	0.993	0.971
	6-way	0.922	0.982	0.979	0.977	0.998	0.977
S9	Med 1 vs. Med 2	0.999	1.0	0.999	1.0	1.0	1.0
	Social vs. mental stress	0.983	0.991	0.988	0.998	0.999	0.998
	6-way	0.975	0.989	0.991	0.996	0.999	0.996
S10	Med 1 vs. Med 2	1.0	1.0	0.999	1.0	1.0	1.0
	Social vs. mental stress	0.944	0.970	0.968	0.983	0.991	0.983
	6-way	0.910	0.961	0.965	0.979	0.995	0.979
S11	Med 1 vs. Med 2	0.999	0.999	0.999	0.999	0.999	0.999
	Social vs. mental stress	0.989	0.989	0.985	0.998	0.999	0.998
	6-way	0.831	0.977	0.982	0.968	0.998	0.968
S13	Med 1 vs. Med 2	1.0	1.0	0.999	1.0	1.0	1.0
	Social vs. mental stress	0.864	0.970	0.973	0.903	0.992	0.903
	6-way	0.909	0.978	0.977	0.976	0.996	0.976
S14	Med 1 vs. Med 2	0.971	0.999	0.999	0.995	0.999	0.999
	Social vs. mental stress	0.963	0.992	0.995	0.972	0.999	0.999
	6-way	0.841	0.967	0.981	0.973	0.992	0.999
S15	Med 1 vs. Med 2	1.0	1.0	1.0	1.0	1.0	1.0
	Social vs. mental stress	0.993	0.998	0.995	0.997	0.999	0.997
	6-way	0.993	0.995	0.995	0.999	0.999	0.999
S16	Med 1 vs. Med 2	0.994	0.996	0.994	0.999	0.999	0.999
	Social vs. mental stress	0.991	0.993	0.993	0.999	0.999	0.999
	6-way	0.926	0.988	0.987	0.987	0.999	0.987
S17	Med 1 vs. Med 2	1.0	1.0	0.999	1.0	1.0	1.0
	Social vs. mental stress	0.941	0.982	0.981	0.973	0.995	0.973
	6-way	0.995	0.979	0.985	0.994	0.997	0.994

## Data Availability

Not applicable.

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
