# Peer review of "Identifying Biomarkers for Accurate Detection of Stress"

_sensors, 2022, doi:10.3390/s22228703_

Round 1
Reviewer 1 Report
This experiment brings a lot to the field of SUD.
Moreover, these devices may bring a lot into methodology of stress assesement.
I find the paper ready for publishing altough here are some of criticism:
- 15 patients is not many. I didnt see a clear distinguishing between control and SUD. Statistical power is probably achieved due to many read-outs timepoints or measurements.
- I would like to see the these 15 SUD patients inclusion criteria more visible.
Reviewer 2 Report
This manuscript deals with the use of sensors to measure physiological signals such as electrodermal activity (EDA). It is a preliminary study. The work is original and within the scope of the Journal and may be an additional source for complement the stress clinical diagnostic.
However, in my opinion there are some points that need to be clarified before it can be published. The authors must consider that sensors for stress monitoring are a very important contribution for wellbeing and a useful complement for clinical diagnosis. However, concerning the SUD it is necessary to have chemical information to have a confident clinical diagnostic and the EDA sensors may help and complement this information. Another possibility it is the EDA use for a preliminary diagnostic test that will need confirmation. As so, the text needs to be revised. Also, the English to be revised. In particular, and as example:
Line 59 Please clarify the sentence “has been introduced through which users …”.
Line 63 to 64 The sentence “are an excellent source to provide continuous and independent recording of a subject’s stress and emotion that cannot be obtained in any other way” is not true, there other ways to obtain subjects stress records namely by electrochemical sensors using biomarkers. The reviewer considers that perhaps the authors are considering the physical measurement with EDA sensors and the sentence needs to be revised.
Line 115 to 116 Please revise the English “the method for how the data was processed before analysis”.
Other Comments by section:
Section 1
In introduction it is not clear how participants are selected for study. For example, they are stressed or subjected for stress but they should not have other pathologies like cystic fibrosis which can give other EDA results.
Section 2.1 Experimental Design – Points that need to be addressed:
1. the selection of subjects or participants needs to be described. For example, which was the control group (not stressed) of participants considered to reference of the results under study.
2. The questionnaires are validated, bibliographic references?
The text needs the English to be revised. In particular:
Section 3. The results are interesting and will be a basis for research community. Please revise the English.
Section 4. Discussion needs English revision. There are other considerations that must be taken in discussion, in particular, sections 4.3 and 4.4 like that physical biomarkers are not the only parameters that need to be used for SUD. It is impossible to be sure about a diagnostic without chemical clinical diagnostic in SUD because patients may be suffering other disorders. EDA may be a complementary and helpful method for preliminary clinical diagnostic.
Section 5. Need to be revised. The authors have to consider that the participants can have other pathologies and by itself the EDA can be also detected in other physiological disorders and not only stress.
Reviewer 3 Report
Dear authors, thank you for presenting an interesting topic that has enormous potential in the future, concerning stress detection by wearable devices.
In your study, you have used the data from 15 subjects to develop biomarkers for stress detection using machine learning algorithms.
Even though the field appears very interesting to me, I have some major issues that have to be resolved prior to publishing.
First of all, I am very confused about the substance abuse disorder and the aim of the study. The whole introduction is about SUD related stress, while the rest of the study is investigating stress in individuals that have nothing to do with SUD. This whole relation has to be additionally explained.
Furthermore, from a clinician's point of view, I have some issues with the subject recruitment. Here again I am not sure whether you simply used public database or were you actually involved in the making of the database. I believe the selection of test subject plays an enormously important role. Indeed, the 15 subjects is not a large number, but nevertheless, if appropriately sampled, could prove valuable. I miss the detailed description of the study group, since it could influence results in a significant way. Potential bias has to be evaluated.
However, the methodology of data analysis is appropriate and machine learning algorithms were well selected with state of art methods.
Round 2
Reviewer 2 Report
I am submitting my revision of the Manuscript ID sensors-1979171, entitled “Identifying Biomarkers for Accurate Detection of Stress”.
This manuscript deals with the use of sensors to measure physiological signals such as electrodermal activity (EDA). It is a preliminary study. The work is original and within the scope of the Journal and may be an additional source for complement the stress clinical diagnostic.
The work is properly presented and discussed. Is original and within the scope of the Journal. After the first revision, the authors have made the corrections suggested by the reviewers. I have no more comments and I consider that the paper is ready for publication.
Author Response
Thank you
Reviewer 3 Report
Dear author, thank you for your responses.
You have managed to clarify several issues, yet they still remain.
If I make summary of your paper review, you have performed machine learning on publicly available dataset to identify biomarkers of stress. Your analytical methods are well performed while the issue remains with the original data you were using. We have only an assumption that the data are actually representative and of decent quality. Since this is the fundamental part of the study it cannot be improved or changed. This way, I expect the detailed description of the issue in the limitation section. We actually know very little about the test subjects and this has to be clearly recognized as a limitation.
